# A non-invasive method to directly quantify surface heterogeneity of porous materials

Wei-Shan Chiang[1,2,3], Daniel Georgi[1], Taner Yildirim[2], Jin-Hong Chen[1] & Yun Liu[2,3]

It is extremely challenging to measure the variation of pore surface properties in complex porous systems even though many porous materials have widely differing pore surface properties at microscopic levels. The surface heterogeneity results in different adsorption/desorption behaviors and storage capacity of guest molecules in pores. Built upon the conventional Porod's law scattering theory applicable mainly to porous materials with relatively homogeneous matrices, here we develop a generalized Porod's scattering law method (GPSLM) to study heterogeneous porous materials and directly obtain the variation of scattering length density (SLD) of pore surfaces. As SLD is a function of the chemical formula and density of the matrix, the non-invasive GPSLM provides a way to probe surface compositional heterogeneity, and can be applied to a wide range of heterogeneous materials especially, but not limited to, porous media and colloids, using either neutron or X-ray scattering techniques.

[1] Aramco Services Company, Aramco Research Center-Houston, Houston, TX 77084, USA. [2] Center for Neutron Research, National Institute of Standards and Technology, Gaithersburg, MD 20899, USA. [3] Department of Chemical and Biomolecular Engineering, University of Delaware, Newark, DE 19716, USA. Correspondence and requests for materials should be addressed to J.-H.C. (email: jinhong.chen@aramcoservices.com) or to Y.L. (email: yunliu@nist.gov)

Surface heterogeneity is ubiquitous in both natural and man-made materials. It represents the coexistence of different chemical and structural properties on the surfaces of a system. The surface heterogeneity significantly influences interactions[1], mechanical properties[2,3], surface reaction rate[4], and storage and transport phenomena[5,6] of the materials. Tuning surface heterogeneity can therefore have wide applications, including in the pharmaceutical industry[7–9], catalysis[10,11], and microfluids[5,6]. Moreover, heterogeneous porous materials have major economic impacts in hydrocarbon extraction, water exploitation, and radioactive waste disposal. The local matrix heterogeneity of coalbed methane and shale reservoirs has been shown to have a large impact on gas adsorption and transportation in pores imbedded in the matrix which determine the ability of natural gas extraction and greenhouse gas sequestration in these systems[12]. Thus there is a great need to understand and quantify the surface properties of these heterogeneous porous materials.

Various techniques have been used to determine the surface heterogeneity of materials, such as atomic force microscopy (AFM)[13], scanning electron microscopy[2,3], energy dispersive spectroscopy (EDS)[2,3], X-ray photoelectron spectroscopy (XPS)[13], auger electron spectroscopy (AES)[14], and secondary ion mass spectrometry (SIMS)[15]. Among these, AFM is useful to obtain surface heterogeneity with the atomic-level resolution. EDS, XPS, AES, and SIMS are powerful to extract the chemical properties on the surface of the materials which directly affect other properties such as biological adhesion[14,15] and mechanical strength[2,3] and are critical for material performance. However, all the techniques mentioned above are, in general, invasive methods and very difficult to be used to probe the surface properties of pores inside porous materials. They usually require careful pre-processing of sample surfaces which may change the original surface properties of the materials. Inverse gas chromatography (IGC)[1,7] is one of the few non-invasive techniques and is useful for characterizing energetic heterogeneity of surfaces, that is, the distribution of surface sites of different energetic levels. However, the results of IGC depend on the probe molecules and assumptions[16,17]. Thus IGC provides information on 'relative' heterogeneity, which can only be used as fingerprints for comparing different materials. Even though the aforementioned surface characterization techniques are powerful and broadly useful, it is still highly desirable to have a new method to characterize the surface heterogeneity that is non-invasive, model-independent, and able to provide compositional properties and statistically reliable values used for comparison between different materials.

In 1951, Porod established the well-known Porod's scattering law for scattering data of materials measured either by X-rays or neutrons[18,19], which is now widely used for extracting the surface area and the average scattering length density (SLD) of materials in various relatively homogeneous two-phase systems such as porous materials[20], biological macromolecules[21], and colloids[22]. SLD is only a function of molecular formula and material density and therefore is an intrinsic property of a material. Despite its wide applications, the conventional Porod's law is, however, intrinsically not applicable to extract the variation of surface properties for heterogeneous systems such as natural rocks, cement pastes, and multi-phase alloys. Owing to the ubiquity of heterogeneity in natural and engineered materials, a new scattering method is needed for obtaining the distribution of the surface properties of the materials.

In this study, we generalize the Porod's scattering law and develop a rigorous new scattering analysis method called generalized Porod's scattering law method (GPSLM) for extracting the surface property variation of heterogeneous systems. Moreover, GPSLM allows to determine the total surface area and surface averaged SLD more accurately than traditional Porod analysis when there is a large variation of pore surface properties. This novel method is non-destructive, model-independent, and applicable to bulk materials. It gives a dimensionless surface heterogeneity parameter that can be used to compare different materials. The obtained surface heterogeneity can be related to the compositional properties of materials through the calculation of SLD.

We first outline our method with the rigorous proof given in Supplementary Note 1. Then, ideal model systems with known heterogeneous surface properties are used to demonstrate the effectiveness of GPSLM. We further apply GPSLM to natural heterogeneous porous materials: kerogens isolated from shale rocks of different maturities. Kerogen is the organic component in shales that cannot be dissolved by any solvents. It is very important to understand the structure, especially the surface properties, of the pores in kerogens because the pores in kerogens are the major storage locations of most produced gas and the pore network controls the matrix flow of gas for production[23–26]. A strong correlation between the surface heterogeneity and maturity of isolated kerogen is observed using GPSLM for the first time.

## Results

**Generalized Porod's scattering law method.** The conventional Porod's scattering law[18,19] states that the asymptotic term of the coherent scattering intensity, $I$, for a two-phase system can be formulated as $I(Q) = 2\pi(\Delta\rho)^2 Q^{-4}\frac{S}{V}$ at high $Q$ when the interface between these two phases is relatively smooth. $Q$ is the scattering wave vector and $\Delta\rho = \rho_1 - \rho_2$ is the contrast of the SLD, where $\rho_1$ and $\rho_2$ are the SLD of phase 1 and phase 2 in a two-phase system, respectively. $S$ and $V$ are the total surface area and total volume seen by a neutron or X-ray beam, respectively. Many experimental systems show clear Porod's law scattering patterns ($I(Q) \propto Q^{-4}$)[21,27–30] that can be measured by either small-angle neutron scattering (SANS) or small-angle X-ray scattering. Even though the conventional Porod's law has been widely applied to porous systems to determine the average SLD and the total surface area, it cannot provide any information of the variation of SLD in pore surfaces that is closely linked to the variation of compositional properties of pore matrix. In fact, it is still very challenging to obtain the distribution of surface properties of pores for heterogeneous porous systems using many existing techniques. Here, we develop the GPSLM that can be a powerful tool to provide essential surface heterogeneity information for heterogeneous materials.

Without losing the generality, a schematic picture of a heterogeneous porous material is shown in Fig. 1. Matrices with different SLDs are shown in different colors. In addition, there are both intra-matrix and inter-matrix pores with different shapes and sizes in the system that can be filled with probing fluids with the SLD as $\rho_f$. The domain of each matrix is associated with its interface, $S$, and $\rho(S)$ is the SLD of the matrix component whose interface is $S$. Four parameters important for materials with heterogeneous surfaces can be extracted with GPSLM: the total surface area ($S_T$), the surface area averaged SLD ($\rho_A$), the surface area averaged second moment of SLD ($\rho_{M^2}$), and the normalized mean square variation of SLD of the matrix ($\Delta_H^2$). The mathematical definitions of $\rho_A$, $\rho_M^2$, and $\Delta_H^2$ are given as follows:

$$\rho_A \equiv \frac{1}{S_T}\int \rho(S)\,\mathrm{d}S, \tag{1}$$

$$\rho_M^2 \equiv \frac{1}{S_T}\int \rho(S)^2\,\mathrm{d}S, \tag{2}$$

$$\Delta_H^2 = \frac{1}{\rho_M^2}\left[\frac{1}{S_T}\int\left(\rho(S)-\rho_A\right)^2\mathrm{d}S\right] = \frac{\rho_M^2 - \rho_A^2}{\rho_M^2}. \tag{3}$$

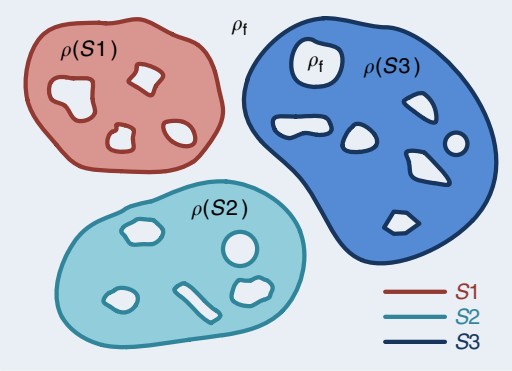

**Fig. 1** Schematic illustration of a heterogeneous porous material. There are both internal and external interfaces, $S$, between the matrices and the pores accessible to the probing fluid in the heterogeneous porous material. The material has surface heterogeneity which represents the coexistence of different chemical and structural properties on the surfaces. Different colors represent matrices with different scattering length densities (SLDs), $\rho(S)$, and the lines depict the interface, $S$. $\rho_f$ is the SLD of probing fluid

It should be noted that integrals in Eqs. (1)–(3) are evaluated over the interface $S$, not the material volume $V$; $\rho_A$, $\rho_M^2$, and $\Delta_H^2$ are surface averaged, not volume averaged parameters.

$\Delta_H$ is called the normalized surface heterogeneity. It characterizes the degree that the SLD of matrix components along the interfaces ($\rho(S)$) deviates from the surface averaged SLD ($\rho_A$). Because SLD is only a function of the chemical formula and material density, $\Delta_H$ can be linked to the surface variation of chemical properties such as stoichiometry and composition. If the chemical formulae of the matrices are known, $\Delta_H$ can be used to estimate the density variation of matrices. Furthermore, $\Delta_H$ is a dimensionless parameter with absolute value and can be used to compare the surface heterogeneity between different materials. Larger $\Delta_H$ means the surface properties over the entire interfaces in the material are more heterogeneous.

The complete mathematical derivation of GPSLM is provided in Supplementary Note 1. Here we only briefly outline the approach. For a system with heterogeneous matrices, we can write the general equation of the Porod's scattering law as

$$I(Q) \xrightarrow{Q \to \infty} 2\pi \langle \Delta\rho_s^2 \rangle Q^{-4} \frac{S_T}{V} = C_{GPS} Q^{-4}. \tag{4}$$

The mean square deviation of SLD (MSD$_{SLD}$), or $\langle \Delta\rho^2 \rangle_s$, in Eq. (4) is defined as $\langle \Delta\rho^2 \rangle_s \equiv \frac{1}{S_T} \int (\rho(S) - \rho_f)^2 \mathrm{d}S$, which integrates through all interface $S$ of all pores in a sample, whose total interfacial area is $S_T$. $C_{GPS} = 2\pi \langle \Delta\rho^2 \rangle_s \frac{S_T}{V}$ is the generalized Porod's scattering constant and it is a function of the SLD of guest fluid in the pores ($\rho_f$).

As aforementioned, for many porous systems, there is a $Q$ region in the scattering pattern following the generalized Porod's law as shown in Eq. (4). The scattering patterns at this $Q$ region change when different probing fluids such as liquid or gas are filled into pores. By varying $\rho_f$, the parameters $\rho_A$, $\rho_M^2$, and $\Delta_H$ can be very straightforwardly determined. Before filling the pores with probing fluid, the pores are empty and $\rho_f = 0$. When loading fluid, $\rho_f$ can be easily tuned by changing either H/D ratio of fluids or pressure (and therefore density) of gases. The scattering contrasts between the fluid and the matrices are changed when tuning $\rho_f$ (Eq. (4)) and the scattering intensity of the material loaded with fluid changes accordingly.

The SANS intensity ratio, which is also the $C_{GPS}$ ratio, at the $Q$ range following the Porod's scattering pattern (Eq. (4)) is defined

as $\mathrm{IR}(Q, \rho_f) \equiv \frac{I(Q, \rho_f)}{I(Q, \rho_f=0)} = \frac{C_{GPS}(\rho_f)}{C_{GPS}(\rho_f=0)} = \mathrm{IR}(\rho_f)$. $\mathrm{IR}(\rho_f)$ is proved to be a parabolic function of $\rho_f$ (Supplementary Note 1). The minimum of $\mathrm{IR}(\rho_f)$ as a function of $\rho_f$ is denoted as $\mathrm{IR}_{min}(\rho_{f,min})$, where $\rho_{f,min}$ is the fluid SLD when $\mathrm{IR}(\rho_f)$ reaches the minimum. Based on the proof in Supplementary Note 1, we have

$$\rho_A = \rho_{f,min}, \tag{5}$$

$$\Delta_H^2 = \mathrm{IR}_{min}(\rho_{f,min}). \tag{6}$$

In the derivation, we assume that the solid matrix SLD does not change when filling the probing fluid in pores, and the $Q$ region is high enough so that the generalized Porod's scattering is due to most of the pore surfaces in a system. Moreover, we assume all the pores are accessible to the guest fluid. Experimentally, it is very straightforward to determine $\rho_{f,min}$ and $\mathrm{IR}_{min}(\rho_{f,min})$ by simply varying $\rho_f$ to find the minimum of $\mathrm{IR}(\rho_f)$. Then by using algebra relations for Eqs. (1)–(6), $\rho_A$, $\rho_M^2$, $\Delta_H$, and $S_T$ can be directly calculated for heterogeneous porous materials. We call this new method the GPSLM. It should be noted that even though the average SLD of various materials has been reported in the literature for decades using the scattering data at the Porod's scattering region, the correct physical meaning of $\rho_{f,min}$ is finally clarified here using the GPSLM, that is, $\rho_{f,min}$ is the surface averaged, rather than volume averaged, SLD of a system. Moreover, GPSLM can be also applied to some systems with highly correlated structure such as core-shell systems (see Supplementary Note 1 and Supplementary Fig. 1 for details).

Comparing to the traditional Porod's scattering law method, two new parameters, $\Delta_H$ and $\rho_M^2$, can be extracted by GPSLM. GPSLM also determines $\rho_A$ and $S_T$ more accurately when $\Delta_H$ is large, and can be used to quantitatively evaluate the effect of the surface heterogeneity on $\rho_A$ and $S_T$ obtained by the traditional Porod analysis which assumes materials are homogeneous (see brief discussion in the Supplementary Note 6).

**Application to theoretical model systems**. In order to test the accuracy of GPSLM, three ideal model systems with known heterogeneous surface properties are constructed with different SLD distribution of the matrices. These model systems are composed of solid spherical particles with different sizes, SLDs, and number densities dispersed in space (details shown in Supplementary Table 1). Pores are formed between spherical particles. The SANS intensity of these systems filled with different pore fluids with various $\rho_f$ can be calculated (details described in Supplementary Note 2). Fig 2a plots the simulated SANS intensity vs. $Q$ for the Model 2 (Supplementary Table 1). As shown in Fig. 2b, $\mathrm{IR}(\rho_f) = \frac{C_{GPS}(\rho_f)}{C_{GPS}(\rho_f=0)}$ is clearly a parabolic function of $\rho_f$. $C_{GPS}$ is obtained from fitting the $Q$ range following the Porod's scattering law with Eq. (4). For different model systems, $\mathrm{IR}(\rho_f)$ reaches different minimum values at different $\rho_f$. $\rho_f$ and $\mathrm{IR}(\rho_f)$ at the minimum of parabolic function, i.e. $\rho_{f,min}$ and $\mathrm{IR}(\rho_{f,min})$, found in Fig. 2b are listed in Supplementary Table 2 together with the theoretical values of $\rho_A$ and $\Delta_H^2$ directly calculated from the exact distribution of SLDs used to construct these models. As expected, $\rho_A = \rho_{f,min}$ and $\Delta_H^2 = \mathrm{IR}_{min}(\rho_{f,min})$ (Supplementary Table 2). GPSLM correctly extracts the surface averaged SLD and surface heterogeneity of SLD from the heterogeneous materials. Moreover, $\Delta_H$ obtained by GPSLM is an absolute value which can be directly compared between different materials.

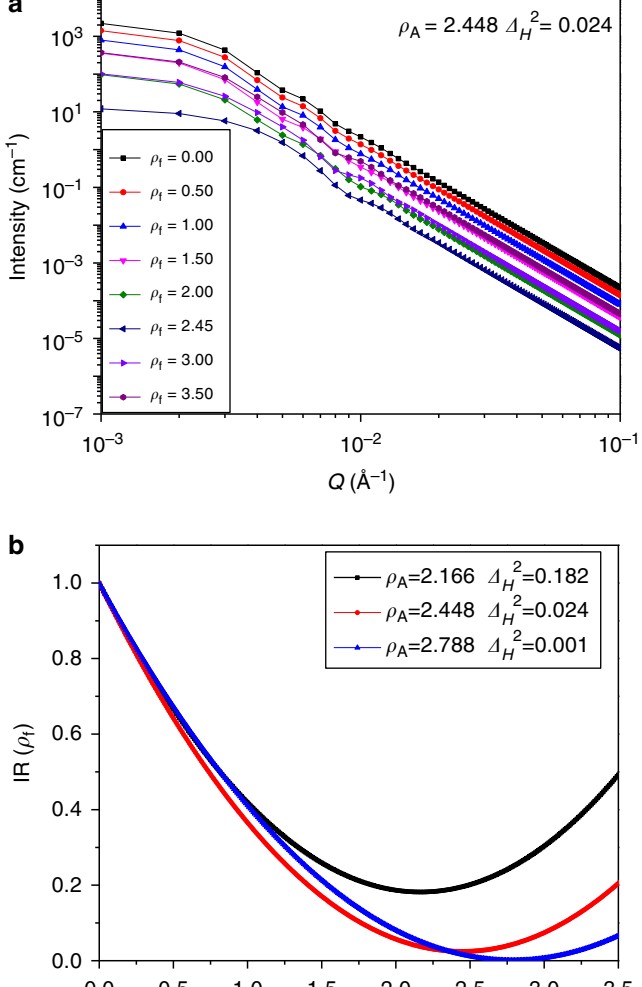

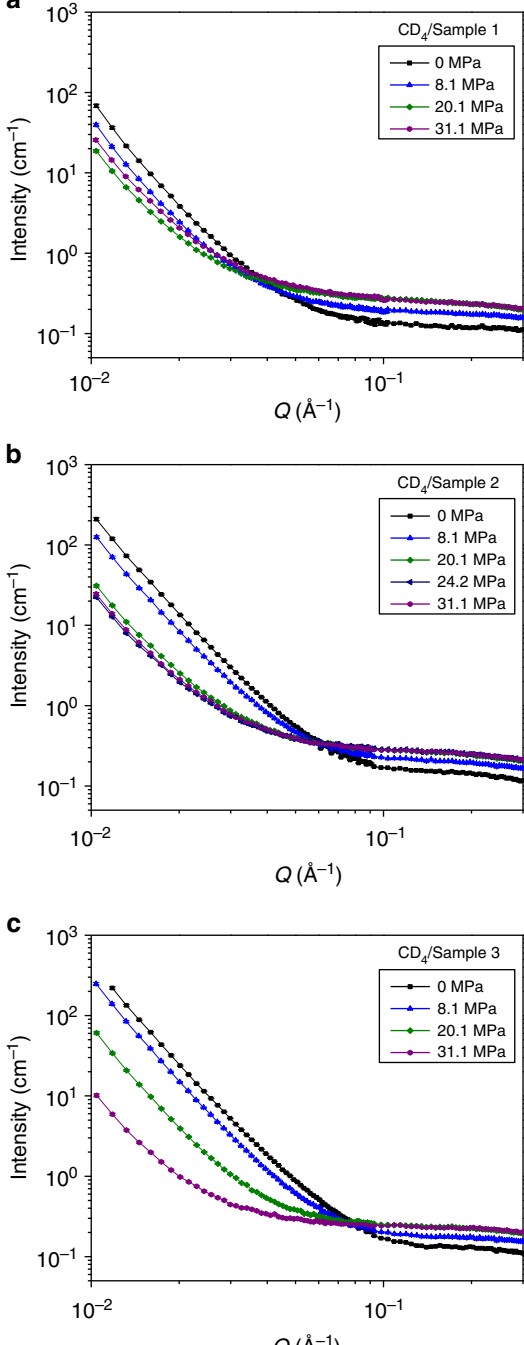

**Fig. 2** Simulated scattering patterns of ideal model systems. **a** Simulation of scattering intensity $I(Q)$ vs. $Q$ for Model 2 ($\rho_A = 2.448$, $\Delta_H^2 = 0.024$) as a function of fluid scattering length density (SLD), $\rho_f$. **b** Intensity ratio, IR ($\rho_f$) = $I(Q, \rho_f)/I(Q, \rho_f = 0) = C_{GPS}(\rho_f)/C_{GPS}(\rho_f = 0)$, vs. $\rho_f$ for Model 1 ($\rho_A = 2.166$, $\Delta_H^2 = 0.182$), Model 2 ($\rho_A = 2.448$, $\Delta_H^2 = 0.024$), and Model 3 ($\rho_A = 2.788$, $\Delta_H^2 = 0.001$). The unit of $\rho_f$ and $\rho_A$ is $10^{10}$ cm$^{-2}$. Details of the model systems and scattering intensity calculation can be found in Supplementary Table 1 and Supplementary Note 2

**Application to shale kerogens**. The GPSLM is applied here to study natural porous materials with heterogeneous surface properties: three kerogen samples isolated from shale rocks using acid digestion[31]. Kerogen is the organic component in shale rocks that is not soluble in any organic solvents. It is commonly agreed that the pores inside kerogen are the major storage locations for light hydrocarbons in most shale reservoirs[25,26]. Therefore, understanding the structure, especially the pore surface properties, of the kerogen in shales is essential to understand the total gas reserve and production rate of a shale gas reservoir.

The BET surface area ($S_{BET}$) and pore volume are first characterized by the commonly used isotherm N$_2$ adsorption at 77 K for the three kerogen samples with different maturities described by vitrinite reflectance ($R_0$) (Supplementary Table 3 and Supplementary Fig. 5). $S_{BET}$, pore volume, and $R_0$ are parameters averaged over the whole samples. Kerogen with higher maturity (higher $R_0$) has slightly higher accessible surface area and pore volume (Supplementary Table 3). However, $S_{BET}$

**Fig. 3** Small-angle neutron scattering experimental data for isolated shale kerogens loaded with probing fluid CD$_4$. **a** Sample 1 (least mature kerogen)/CD$_4$. **b** Sample 2/CD$_4$. **c** Sample 3 (most mature kerogen)/CD$_4$. CD$_4$ pressure ranges from 0 to 31.1 MPa. At high $Q$, the intensity flattens off due to the incoherent scattering background contributed from both kerogen and CD$_4$. Error bars represent one standard deviation. More small-angle neutron scattering data with different CD$_4$ pressures are available (Supplementary Figs 3b–d). Details of the kerogens can be found in Supplementary Table 3

and pore volume are similar among the three kerogens even though their $R_0$ values are very different. Thus, other quantities are needed for better sample characterization.

To apply the GPSLM, gas is used as the probing fluid as an example and different $\rho_f$ is achieved by changing the gas pressure. We conduct SANS on the three kerogen samples loaded with

deuterated methane $CD_4$ at different pressures ranging from 0 to 31.1 MPa (Fig. 3 and Supplementary Figs 3b–d). The scattering intensity at the low $Q$ region follows the generalized Porod's scattering with $Q^{-4}$ power law feature (Fig. 3 and Supplementary Fig. 3) and the solid matrix SLD is not expected to be significantly affected by the loading gas (see Supplementary Note 4). Thus the GPSLM can be applied. The density of methane inside kerogen pores at room temperature can be assumed to be bulk methane density in our cases[32–34], i.e. $\rho_f = \rho_{CD_4}$ (details are given in Supplementary Note 4). $\rho_{CD_4}$ is SLD of bulk $CD_4$ and is calculated and plotted as a function of pressure in Supplementary Figure 2.

Separate SANS measurements of these kerogens loaded with helium demonstrate that the structure of solid kerogen matrix does not change in the pressure range being studied (Supplementary Fig. 3a and Supplementary Note 3). Therefore, the change of SANS curves for kerogens loaded with $CD_4$ (Fig. 3) is due to the density change of $CD_4$ in the samples as indicated by the Eq. (4). When increasing $CD_4$ pressure, $\rho_{CD_4}$ increases while the SLDs of the kerogen matrices maintain as constant positive values that depend on the maturity of the kerogen[35]. The SANS intensity, which is proportional to $\langle \Delta \rho^2 \rangle_s = \frac{1}{S_T} \int \left( \rho(S) - \rho_{CD_4} \right)^2 dS$, decreases with $\rho_{CD_4}$ first and then increases when $\rho_{CD_4}$ becomes larger than the surface area averaged SLD of the matrices, $\rho_A$ (Fig. 3 and Supplementary Fig. 3).

The intensity ratios, which are also the ratios of generalized Porod's constant $IR(\rho_f) = IR(\rho_{CD_4}) = \frac{C_{GPS}(\rho_f)}{C_{(GPS)}(\rho_f = 0)}$, obtained at different $CD_4$ pressures for the three kerogens are plotted as a function of $\rho_{CD_4}$ (Fig. 4a). $C_{GPS}$ is extracted from fitting the SANS data in the $Q$ range following generalized Porod's scattering using Eq. (4). For less mature samples, i.e. Sample 1 and Sample 2, the pressure range of $CD_4$ being used (0 to 31.1 MPa) allows $IR(\rho_f)$ to reach the minimum, i.e. $IR_{min}(\rho_{f,min})$, and eventually $IR(\rho_f)$ increases again when the pressure further increases. For the most mature sample, i.e. Sample 3, the current pressure range is not high enough to reach $IR_{min}(\rho_{f,min})$ (Fig. 4a). However, since $IR(\rho_f)$ measured at the highest pressure (31.1 MPa) for Sample 3 is very close to zero, we expect that this value is close to $IR_{min}(\rho_{f,min})$ and is the upper limit of $IR_{min}(\rho_{f,min})$. Equation (5) indicates the fluid SLD, $\rho_f = \rho_{CD_4}$, at minimum $IR(\rho_f)$ is equal to the surface average of kerogen SLD, $\rho_A$. More mature kerogen is shown to have higher $\rho_A$ (Fig. 4a) that is consistent with the hydrogen carbon ratio obtained with the Prompt Gamma-ray Activation Analysis (PGAA) shown in Supplementary Table 5 and Supplementary Note 7. This indicates that less hydrogen is in the sample at this stage of maturation, which is consistent with the literature[35]. Using Eq. (6), the normalized surface heterogeneity, $\Delta_H$, is calculated for the kerogens and plotted as a function of vitrinite reflectance, $R_0$ (Fig. 4b). The parameters extracted from GPSLM are listed in Supplementary Table 4. Our results for the first time demonstrate the direct experimental observation of the decrease in $\Delta_H$ with the increase in $R_0$ for the isolated kerogens by acid digestion, suggesting that the surface properties of the shale kerogen become more homogeneous during the maturation.

After obtaining $\rho_A$ and $\Delta_H$, the total interfacial area, $S_T$, can be easily calculated using Eq. (4) (see details in Supplementary Note 1 and Supplementary Note 5). The specific surface area, i.e. the total interfacial area per mass of dry kerogen, obtained from the GPSLM, $S_{GPS}$, is compared with the BET surface area measured by isotherm $N_2$ adsorption at 77 K, $S_{BET}$ (Supplementary Table 3). Both $S_{GPS}$ and $S_{BET}$ have the same trend that

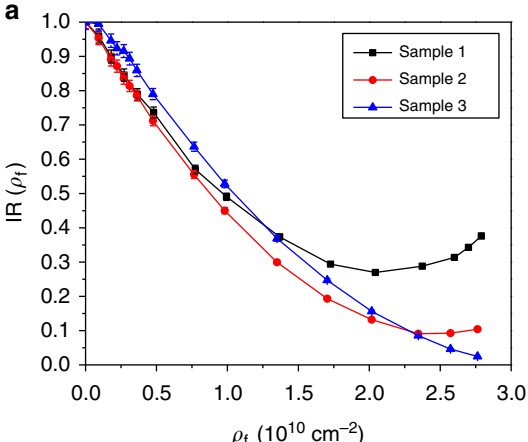

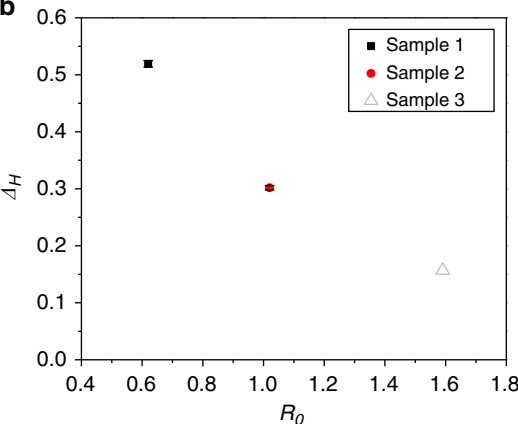

**Fig. 4** Analysis of shale kerogens using generalized Porod's scattering law method. **a** Intensity ratio, $IR(\rho_f) = I(Q, \rho_f)/I(Q, \rho_f = 0)$, which is also the ratio of generalized Porod's scattering constant $C_{GPS}(\rho_f)/C_{GPS}(\rho_f = 0)$, for Sample 1 (black squares, least mature kerogen), Sample 2 (red circles), and Sample 3 (blue triangles, most mature kerogen) as a function of $\rho_f = \rho_{CD_4}$. **b** Normalized surface heterogeneity $\Delta_H$ of the shale kerogens as a function of vitrinite reflectance ($R_0$). The highest methane pressure available in current experiment setup cannot reach the minimum of $IR(\rho_f)$ for Sample 3 so the approximate $\Delta_H$ based on the highest pressure measurement is shown (gray open triangle) for Sample 3

kerogen with higher maturity has higher specific surface area. Since the SANS data following the Porod's law region from about 0.012 Å$^{-1}$ to 0.03 Å$^{-1}$ are used to calculate $S_{GPS}$, only the surface area of pores with length scale approximately larger than $L \approx \frac{2\pi}{Q} \approx$ 200 Å is included. Because $S_{GPS}$ is found to be very close to $S_{BET}$ for all the three kerogens (Supplementary Table 3), this indicates that most of the surface area measured by $N_2$ adsorption is contributed by large pores with size approximately larger than 200 Å and there are very few small pores in these kerogens.

For shales, the surface properties of pores are keys for determining the storage capacity and distribution of hydrocarbons, especially when hydrocarbon mixtures are considered[23]. In the literature, vitrinite reflectance ($R_0$) is often used as an index for quantifying the maturity of kerogen[36]. Other indices such as total organic carbon, hydrogen index, oxygen index, etc., are also used to characterize the quality and types of kerogen[37]. However, these indices are all average values for the samples and the variation of kerogen surface properties is difficult to obtain. The new GPSLM allows one to obtain $\Delta_H$ (the width of distribution) and therefore provides essential details for the materials.

## Discussion

The classic Porod's scattering law has been widely applied for extracting the surface-to-volume ratio in two-phase systems such as porous materials[20] and colloids[22]. The GPSLM developed in this work significantly extends the applications of original Porod's law from simple homogeneous systems to heterogeneous multiple-phase systems. The new GPSLM is based on the widely used contrast variation technique but is able to extract much more information such as the normalized surface heterogeneity ($\Delta_H$), which is the deviation of SLD from the surface averaged SLD ($\rho_A$). $\Delta_H$ directly links to the distribution of compositional properties in the materials. GPSLM can also obtain the values of the total surface area ($S_T$) and $\rho_A$ more accurately in systems with a large variation of surface properties than simply using the traditional Porod's analysis by assuming homogeneous materials. For systems with heterogeneous surface properties, the accuracy of $S_T$ and $\rho_A$ determined by the traditional Porod's analysis depends on the value of $\Delta_H$ and the way the contrast variation method is conducted, which is discussed in details in Supplementary Note 6 and Supplementary Fig. 4.

It is worth noting that the surface averaged properties are more useful than the volume averaged properties in systems involved in reactions on the surface, such as gas adsorption[12,38], biological target[8,9], and catalysis[10,11], because the distribution of the surface properties can significantly influence the performance of materials. By using the novel GPSLM, we experimentally quantify surface heterogeneity of multiple isolated shale kerogen samples, and discover the interesting correlation between surface heterogeneity and kerogen maturity. To the best of our knowledge, this is the first time that quantitative values of the normalized surface heterogeneity, $\Delta_H$, are extracted through scattering method.

In addition to heterogeneous porous materials, GPSLM can be easily applied to other systems to extract $\rho_A$, $\Delta_H$, and $S_T$ of the heterogeneous surface when the guest fluid SLD can be tuned. For colloidal and some biological suspensions where particles are dispersed in continuous medium such as water or other solvents, the fluid SLD can be tuned by mixing hydrogenated and deuterated solvents, such as $H_2O/D_2O$ or $C_2H_5OH/C_2D_5OH$, with different ratios. For hydrophobic porous materials whose pores are not accessible to $H_2O/D_2O$, or other solids whose surface properties needed to be characterized without introducing liquid solvents, gas loading with different pressures can be used.

## Methods

**Acid digestion for shale kerogen isolation**. Approximately 20 g of each shale sample was ground to pass a 20 mesh sieve. The samples were then extracted with dichloromethane to remove any soluble hydrocarbons. After drying, the samples were treated with hydrochloric acid (HCl) and left to stand in HCl for at least 2 h with occasional stirring. The samples were rinsed four times with water to remove any calcium or magnesium ions that were released by the HCl.

The decarbonated samples were treated with 70% hydrofluoric acid (HF). The acid was added slowly with stirring until there was no more reaction. After the samples were cooled to room temperature, the samples were transferred to a centrifuge tube, centrifuged, and fresh HF was added to tube. Samples were then placed in an ultrasonic and heated for at least 3 h at 50 °C. Samples stayed in a tube with occasional shaking for at least 2 days.

Samples were rinsed three times and then concentrated HCl was added and samples were heated for at least 3 h at 50 °C. Samples were rinsed twice with water and twice with distilled water and were ready for freeze drying.

**Small-angle neutron scattering**. SANS measurements were performed at nSoft-10m SANS and NGB-30 m SANS at the National Institute of Standards and Technology (NIST) Center of Neutron Research (NCNR). The incident neutron wavelength, $\lambda$, was chosen to be 5 or 6 Å and the sample-to-detector distances, SSDs, were selected to cover a scattering vector ($Q$) range from 0.0014 to 0.568 Å$^{-1}$. All SANS data were corrected for the sample transmission, the background scattering, and the detector sensitivity to obtain the absolute intensity based on a standard procedure described elsewhere[39].

The kerogen samples were degassed under vacuum for 2 days before the SANS measurement. The degassed samples were loaded into the high pressure (HP) cells

in the helium-filled glovebox. The HP cells are made of stainless steel and contain sapphire windows. The HP cells only allow the reliable data up to $Q = 0.3$ Å$^{-1}$. Helium (He) and deuterated methane ($CD_4$) pressure are controlled by a 100HLf hazardous location syringe pump with a gas loading line connecting to the syringe pump. SANS measurement was first conducted on the shale kerogen under vacuum using a turbo pump linked to the syringe pump. He with 31.1 MPa was then loaded into the kerogen for in situ SANS measurement. After the He measurement, the He was pumped out by the turbo pump. $CD_4$ with different pressures was loaded into the samples to perform in situ SANS study. All the measurements were maintained at 21 °C by the cooling bath system. $CD_4$ was purchased from Cambridge Isotope Laboratories, Inc. Thermodynamic properties of the bulk methane were calculated using NIST standard reference database software REFPROP[40].

**Volumetric isotherm gas adsorption**. Isotherm gas sorption measurements were performed on a carefully calibrated and high accuracy Sieverts apparatus under computer control. Instrument and measurement-protocol details have been published elsewhere[41].

**Prompt gamma-ray activation analysis**. Cold neutron prompt gamma-ray activation analysis (PGAA) was performed at the NGD beamline at NCNR. The details of the facilities can be found elsewhere[42]. The samples were vacuumed for two days prior to the measurement to remove residual moisture..

**Data availability**. All relevant data are available from the authors upon request.

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

## Acknowledgements

The authors thank Ronald Jones, Kathleen Weigandt, Alan Ye, and Juscelino Leao for the *n*Soft beamline support and thank Danyal Turkoglu and Rick Paul for the PGAA beamline support. The authors also thank David Jacobi for arranging isolation of kerogen samples from shales and measurement of the maturity. Y.L. acknowledges the partial support of cooperative agreements 70NANB12H239 and 70NANB10H256 from NIST, U.S. Department of Commerce. This work was funded in part by Aramco Services Company and access to NGB-30 m SANS was provided by the Center for High Resolution Neutron Scattering, a partnership between the National Institute of Standards and Technology and the National Science Foundation under Agreement No. DMR-1508249. Identification of a commercial product does not imply recommendation or endorsement by the National Institute of Standards and Technology, nor does it imply that the product is necessarily the best for the stated purpose.

## Author contributions

Y.L. and J.H.C. led the project. W.S.C. and Y.L. conceived the method and performed SANS experiments. W.S.C. performed the data analysis. D.G. and J.H.C. provided sample information and geological knowledge. T.Y. performed the $N_2$ isotherm gas adsorption. All authors contributed to the manuscript writing.

## Additional information

**Competing interests:** The authors declare no competing financial interests.

