## [Peer Review File · Nature Communications]

Reviewers' comments:

Reviewer #1 (Remarks to the Author):

The paper is an excellent work about the surface area surface heterogeneity determination of solid adsorbent materials if we have 3 different phases. In this sense it is a new contribution on the topics of adsorption and surface characterization and it could be very useful for characterization of porous materials for catalyst but crude oil reservoir samples as composite materials on nanoscale. The presentation of the calculation method and the experimental data are detailed. The list of references is informative as well.

The referee would like to ask an independent experiment for surface heterogeneity measurement for example calorimetric data or adsorption isotherms parameters (or constant C from the BET equation, high or lower surface energy) of adsorption at solid/gas interface.

Using these data the very important heterogeneity parameter (H) would be better validated.

Reviewer #2 (Remarks to the Author):

Chiang et al reported a scattering method to determine the chemical heterogeneity of porous materials. Studying the chemical heterogeneity of polymeric complex materials like kerogens could be of interest but challenging. Authors' method will certainly be useful for the related field. The method and outcome are significant, but they are not so novel to draw attentions of general audience. Detailed critics are following:

1. This method is basically a contrast variation method that is very common in SANS or SAXS experiment. With some exaggeration, so common that every SANS works include some types of contrast variation measurements. Various analysis formula are out there for specific cases and some for more general ones. This manuscript, however, did not discuss enough how novel their work is in those regards, considering that the method development is a key in the work.
2. Accuracy of the method is also somewhat arguable. For example, by definition, this method only works when the concentration of kerogen is very small. But in reality it cannot be. More generally, the SANS contrast should be $\rho(r) - \text{mean}(\rho(r))$ instead of $\rho(r) - \rho_{\text{fluid}}$.
3. Analyzing data for one Q value is not so satisfactory. It would be better to use as many Q points as possible or use the slope instead of a point.
4. Related to 2, this method becomes less useful if a part of a sample has both chemical and structural heterogeneity distinctive to other parts. For example, let's say the domain 1 in Figure 1 has a fractal exponent 3 and the others have 4. If the concentration of macro pores (pores between particles) is not so much different from particles, this will affect the contrast a lot too. See Spalla et al, J. Appl. Crystallogr. 2003, 36, 338–347. or Chavez Panduro et al. J. Appl. Crystallogr. 2012, 45, 881–889.
5. Variation of chemical heterogeneity of kerogens as a function of RO is a good result, which however is well within expectations made by measurements and studies. Authors may want to be more quantitative, for example how much hydrogen atoms can there be.

Reviewer #3 (Remarks to the Author):

The contribution claimed in this paper is the modification of Porod's scattering law for heterogeneous materials. The authors fail to demonstrate the improvement over the simple application of the scattering law for homogeneous materials. How do their calculations compare with the standard Porod's law calculation on the same data set? Previous studies by Thomas et al. 2014 applying the homogeneous law show results which are very similar and more importantly indicate that the acid digestion process used to isolate the kerogen studied alters the properties of the kerogen.

The author's interpretation of BET data is simplistic as we now know that sample preparation drastically affects the surface area measured in N₂ adsorption. Furthermore, inversion for pore

dimensions typically requires the assumption of slits which is: 1) not true and 2) inconsistent with typical scattering models.

The authors make a claim in the discussion section that by using "GPSLM" they can quantify surface heterogeneity of kerogen in shale but as Thomas et al. (2014) claim, they may not be seeing the true kerogen response but the variability of the kerogen reaction to the acid digestion process.

Thomas, J. J., J. J. Valenza II, P. R. Craddock, K. D. Bake and A. E. Pomerantz, 2014. The neutron scattering

Responses to Reviewers' comments:

For Reviewer #1

Comments: The paper is an excellent work about the surface area surface heterogeneity determination of solid adsorbent materials if we have 3 different phase. In this sense it is a new contribution on the topics of adsorption and surface characterization and it could be very usefully for characterization of porous materials for catalyst but crude oil reservoir samples of composite materials on nanoscale. The presentation of the calculation method and the experimental data are detailed. The list of references is informative as well. The referee would like to ask an independent experiment for surface heterogeneity measurement for example calorimetric data or adsorption isotherms parameters (or constant C from the BET equation, high or lower surface energy) of adsorption at solid/gas interface. Using these data the very important heterogeneity parameter (H) would be better validated.

Responses: the authors thank the reviewer for acknowledging that the generalized Porod's scattering law method (GPSLM) is a new contribution on the topics of adsorption and surface characterization and GPSLM is an useful tool for a variety of fields.

As required by the reviewer, the BET fitting results together with the BET constant C are now put into Supplementary Figure 4 and Supplementary Table 3. It should be noted the surface heterogeneity measured here is related to compositional distribution of the kerogens which is not necessarily related to the surface energy distribution of the materials.

Reviewer #2

Comment: Chiang et al reported a scattering method to determine the chemical heterogeneity of porous materials. Studying the chemical heterogeneity of polymeric complex materials like kerogens could be of interest but challenging. Authors' method will certainly be useful for the related field.

Responses: We acknowledge the reviewer's appreciation that the generalized Porod's scattering law method (GPSLM) is useful for related fields.

Comments: The method and outcome are significant, but they are not so novel to draw attentions of general audience.

Responses: We are pleased that the reviewer agrees that the method and that outcomes are significant. And we believe that the reviewer's concerns for the novelty of this work are due to some misunderstanding of the literature results. We will address the questions raised by this reviewer below. The manuscript is modified accordingly.

Comments: Detailed critics are following:

1. This method is basically a contrast variation method that is very common in SANS or SAXS experiment. With some exaggeration, so common that every SANS works include some types of contrast variation measurements. Various analysis formula are out there for specific cases and some for more general ones. This manuscript, however, did not discuss enough how novel their work is in those regards, considering that the method development is a key in the work.

Responses: The wide use of the contrast variation method in fact shows the importance of the GPSLM. It has been decades that researchers have used contrast variation to extract the “average” scattering length density (SLD) of dispersed scattering objects in the samples. However, no one has been able to extract the surface heterogeneity information from the widely used contrast variation method. In fact, obtaining the surface heterogeneity of porous materials quantitatively is a challenging task for many other techniques too. Our GPSLM is an easy and straightforward method to determine quantitative surface heterogeneity. The fact that the contrast variation has been widely used in many small-angle neutron scattering (SANS) and small-angle X-ray scattering (SAXS) experiments indicate that the GPSLM can really draw the attention of general audience. By applying GPSLM, researchers still use the same contrast variation experiment but now can get novel information out of the experimental scattering data such as the distribution of SLD on the interface. Moreover, GPSLM can extract the specific surface area and average SLD more accurately than the traditional method which assumes the systems to be homogenous. The accuracy of the traditional method depends on the conditions and experimental values being used and is now discussed in the supplementary information of the revised manuscript.

In addition, for several decades there has been no clear conclusion as to what the obtained average SLD of the samples from contrast variation is for porous materials. It has been assumed as “volume-averaged” SLD by many people. GPSLM demonstrates for the first time that it is actually the “surface-averaged” SLD of scattering objects. GPSLM provides the correct physical meaning of the “average SLD” being found in regular contrast variation.

We have modified the “Discussion” section in the manuscript in order to better address the importance and novelty of GPSLM.

Comments: 2. Accuracy of the method is also somewhat arguable. For example, by definition, this method only works when the concentration of kerogen is very small. But in reality it cannot be. More generally, the SANS contrast should be $\rho(r) - \text{mean}(\rho(r))$ instead of $\rho(r) - \rho_{\text{fluid}}$.

Responses: We respectfully disagree with the reviewer. As long as the Porod’s law scattering region

is due to the surface of most pores including the pores inside kerogen grains, our method does NOT depend on the kerogen concentration. It is based on the original Porod's scattering law for two-phase system and the derivation does not involve any assumption for concentration, as can be seen in the original papers from Porod (Porod, G. Die Röntgenkleinwinkelstreuung von dichtgepackten kolloiden Systemen. *Kolloid-Zeitschrift* **125**, 51–57 (1952); Porod, G. Die Röntgenkleinwinkelstreuung von dichtgepackten kolloiden Systemen. *Kolloid-Zeitschrift* **124**, 83–114 (1951)). Therefore, it is not a problem for high grain concentration in kerogen. (In the derivation, there is no need to worry about the so-called inter-particle structure factor, $S(Q)$, if the reviewer thinks about this from the scattering of colloidal type systems, where $I(Q)=n*P(Q)*S(Q)$. At high Q where the Porod's scattering shows up, $S(Q)$ is about one. This is especially true if the systems are highly disordered, which is the case for our kerogen samples.)

The expression of small-angle scattering intensity can be written in different ways which will result in slightly different forms of writing the contrast term, including “ $\rho(r) - \text{mean}(\rho(r))$ ” and “ $\rho(r) - \rho_{\text{fluid}}$ ” as pointed out by the reviewer. However, the difference in the contrast terms will result in different ways to calculate the scattering intensity when treating both the matrix and pores filled with fluids. But both expressions will give the same result if the calculations are treated correctly. More details can be found in many books such as Glatter, O. & Kratky, O. *Small angle x-ray scattering*. (Academic Press, 1982); Roe, R. J. *Methods of X-ray and neutron scattering in polymer science*. (Oxford University Press, 2000). We chose to express the contrast term as “ $\rho(r) - \rho_{\text{fluid}}$ ” as at the high Q region, the scattering contrast of the Porod's law scattering is only due to the contrast between fluids in pores and the solid matrix. If using $\rho(r) - \text{mean}(\rho(r))$, we will need to calculate the scattering intensity from the matrix as $\rho(r)_{\text{matrix}} - \text{mean}(\rho(r))$ cannot be ignored. But both expressions lead to the same scattering results for $Q > 0$. If requested, we can provide the full mathematical derivation for it in the Supporting Information of this paper.

Comments: 3. Analyzing data for one Q value is not so satisfactory. It would be better to use as many Q points as possible or use the slope instead of a point.

Responses: We agree with and thank the reviewer for raising this excellent point. In the revised version of the manuscript, we have used multiple Q points as suggested by the reviewer by fitting the Porod's scattering Q range. It should be noted that this slight change does not alter the novelty of our method. The new results agree with the results obtained in the previous version of the manuscript.

Comments: 4. Related to 2, this method becomes less useful if a part of a sample has both chemical and structural heterogeneity distinctive to other parts. For example, let's say the domain 1 in Figure 1 has a fractal exponent 3 and the others have 4. If the concentration of macro pores (pores between particles) is not so much different from particles, this will affect the contrast a lot too. See Spalla et al, J. Appl. Crystallogr. 2003, 36, 338–347. or Chavez Panduro et al. J. Appl. Crystallogr.

Responses: It seems to us that this reviewer is concerned that not all porous materials have the Porod's law scattering region with the Q^{-4} dependence. We agree with the reviewer that our method has its limitation as not all samples measured by small-angle scattering show generalized Porod's scattering, i.e. $I(Q) \sim c * Q^{-4}$. Similar to any experimental method for any research, no method can address all research problems. However, we would like to point out Porod's scattering is found in many different types of materials, ranging from silica materials to alloys. GPSLM is therefore broadly useful because many materials, even if not all, in different fields produce Porod's scattering pattern at high Q . In addition to the application to the hugely important geological samples in current study, a few more examples/fields are given here:

- Silica Materials: Silicate Glass (Cailleateau, C. *et al.* Insight into silicate-glass corrosion mechanisms. *Nat. Mater.* **7**, 978–983 (2008)); Silica Aerogel (Schaefer, D. W. & Keefer, K. D. Structure of Random Porous Materials: Silica Aerogel. *Phys. Rev. Lett.* **56**, 2199–2202 (1986)); Silica Nanoparticles (Hendrik K. Kammler, Gregory Beaucage, †, Roger Mueller, A. & Pratsinis*, S. E. Structure of Flame-Made Silica Nanoparticles by Ultra-Small-Angle X-ray Scattering. *Langmuir* **20**, 1915–1921 (2004)).
- Energy Materials: Organic Photovoltaics (Yin, W. & Dadmun, M. A New Model for the Morphology of P3HT/PCBM Organic Photovoltaics from Small-Angle Neutron Scattering: Rivers and Streams. *ACS Nano* **5**, 4756–4768 (2011)); Metal–Organic Framework (Tsao, C.-S. *et al.* Characterization of Pore Structure in Metal–Organic Framework by Small-Angle X-ray Scattering. *J. Am. Chem. Soc.* **129**, 15997–16004 (2007)).
- Biological Materials: Macromolecules (Rambo, R. P. & Tainer, J. A. Characterizing flexible and intrinsically unstructured biological macromolecules by SAS using the Porod-Debye law. *Biopolymers* **95**, 559–571 (2011)); Bone (Fratzl, P., Fratzl-Zelman, N., Klaushofer, K., Vogl, G. & Koller, K. Nucleation and growth of mineral crystals in bone studied by small-angle X-ray scattering. *Calcif. Tissue Int.* **48**, 407–413 (1991)); Dental Nanocomposites (Wilson, K. S., Allen, A. J., Washburn, N. R. & Antonucci, J. M. Interphase effects in dental nanocomposites investigated by small-angle neutron scattering. *J. Biomed. Mater. Res. Part A* **81A**, 113–123 (2007)).
- Coatings: Ceramics coatings (Kulkarni, A. *et al.* Studies of the microstructure and properties of dense ceramic coatings produced by high-velocity oxygen-fuel combustion spraying. *Mater. Sci. Eng. A* **369**, 124–137 (2004)); Industrial Thermal Barrier Coating (Kulkarni, A. A. *et al.* Microstructure-Property Correlations in Industrial Thermal Barrier Coatings. *J. Am. Ceram. Soc.* **87**, 1294–1300 (2004)).

- Microgels: Temperature Sensitive Microgel Colloids (Stieger, M., Richtering, W., Pedersen, J. S. & Lindner, P. Small-angle neutron scattering study of structural changes in temperature sensitive microgel colloids. *J. Chem. Phys.* **120**, 6197–6206 (2004)); Responsive Polymer Colloids to Biomaterials (Saunders, B. R. *et al.* Microgels: From responsive polymer colloids to biomaterials. *Adv. Colloid Interface Sci.* **147–148**, 251–262 (2009)).
- Microemulsions: Water-in-CO₂ Microemulsions (Eastoe, J. *et al.* Water-in-CO₂ Microemulsions Studied by Small-Angle Neutron Scattering. *Langmuir* **13**, 6980–6984 (1997)); Water-in-Heptane Microemulsion (Bumajdad, A. *et al.* Compositions of Mixed Surfactant Layers in Microemulsions Determined by Small-Angle Neutron Scattering. *Langmuir* **19**, 2560–2567 (2003)).
- Construction Materials: Cement (Thomas, J. ., Jennings, H. . & Allen, A. . The surface area of cement paste as measured by neutron scattering: evidence for two C-S-H morphologies. *Cem. Concr. Res.* **28**, 897–905 (1998)).
- Alloys: Zhong, S. Y. *et al.* Study of the thermal stability of nanoparticle distributions in an oxide dispersion strengthened (ODS) ferritic alloys. *J. Nucl. Mater.* **428**, 154–159 (2012); Heintze, C., Bergner, F., Ulbricht, A. & Eckerlebe, H. The microstructure of neutron-irradiated Fe–Cr alloys: A small-angle neutron scattering study. *J. Nucl. Mater.* **409**, 106–111 (2011).

These works mentioned above have all been heavily cited in their corresponding fields. The ubiquity of the Porod's scattering pattern again demonstrates that GPSLM can be applied to a wide range of fields and can draw general interest.

The cited papers by this reviewer discussed some special cases where some materials may show two Porod's scattering law regions. Here, our GPSLM also works in these cases if Q is large enough so that the scattering of surfaces is due to pores including both macro-pores and pores inside grains.

In addition, as mentioned in the response to the Comment 2 of the reviewer, GPSLM does not depend on the concentration of macropores because it is based on the original Porod's scattering law whose derivation does not involve any assumption for concentration.

We have revised the manuscript accordingly to address the aforementioned questions raised by this reviewer.

Comments: 5. Variation of chemical heterogeneity of kerogens as a function of R0 is a good result, which however is well within expectations made by measurements and studies. Authors may want to be more quantitative, for example how much hydrogen atoms can there be.

Responses: We gratefully acknowledge for the reviewer’s appreciation of the measurement of the surface heterogeneity. To the best of our knowledge, we are not aware of any report so far to successfully extract this experimental information even though the results generally agree with what industry experts expect.

The hydrogen atom density (hydrogen index) is related to R_0 . We have done a separate Prompt Gamma-ray Neutron Activation Analysis (PGAA) experiment to extract hydrogen to carbon ratio (H/C) of the kerogen that is fully consistent with our results obtained by GPSLM method. H/C is decreased with R_0 as expected and agrees with the results reported by **Thomas *et al*** (Thomas, J. J., Valenza, J. J., Craddock, P. R., Bake, K. D. & Pomerantz, A. E. The neutron scattering length density of kerogen and coal as determined by CH₃OH/CD₃OH exchange. *Fuel* **117**, 801–808 (2014)). The Table below listed the H/C ratio we determine from PGAA and is now included in the revised manuscript:

Sample ID	R_0 (%)	H/C
Sample 1	0.62	1.02(4)
Sample 2	1.02	0.93(4)
Sample 3	1.59	0.63(3)

Reviewer #3

Comments: The contribution claimed in this paper is the modification of Porod’s scattering law for heterogeneous materials. The authors fail to demonstrate the improvement over the simple application of the scattering law for homogeneous materials. How do their calculations compare with the standard Porod’s law calculation on the same data set?

Responses: The key improvement is to probe the heterogeneity of the surface scattering length density (SLD) in heterogeneous materials. The standard Porod’s law cannot extract any quantitative information of the surface heterogeneity. In fact, there are very few, if not any, methods that could provide the surface heterogeneity information for porous materials.

Overall, the generalized Porod’s scattering law method (GPSLM) can provide information for four parameters: the total surface area (S_T), the surface area averaged SLD (ρ_A), the surface area averaged second moment of SLD (ρ_M^2), and the normalized mean square variation of SLD of the matrix (Δ_H^2 , or the surface heterogeneity). Among the four parameters, the traditional Porod’s law method (homogeneous analysis) can only estimate S_T and ρ_A . However, when there is surface

heterogeneity, the estimated S_T and ρ_A are influenced by Δ_H^2 .

We have estimated the surface heterogeneity effect on S_T and ρ_A obtained by the traditional Porod's method when there is a non-zero Δ_H^2 , i.e. there are heterogeneous surfaces. In general, when Δ_H^2 is small or the guest fluid SLD (ρ_f) is far away from ρ_A , i.e. $|\rho_f - \rho_A|$ is very large, the expression for the normalized scattering intensity of heterogeneous materials reduces to the homogeneous case and therefore the estimated ρ_A from traditional Porod's method is very close to the real ρ_A obtained from GPSLM. But traditional Porod's method still either overestimates or underestimates S_T depending on the parameters used. The accuracy of S_T and ρ_A obtained by the traditional Porod's method for kerogen Sample 1 (kerogen with the lowest maturity and highest heterogeneity in our study) is now discussed in the Supplementary Note 6 and Supplementary Figure 3 in the revised supplementary document.

In addition, it has been decades that researchers have used the contrast variation result at the Porod's law scattering region to obtain "the average SLD". However, it has never been clear if this is volume average, which is sometimes commonly assumed, or the surface average of SLD. The derivation of GPSLM gives the correct and clear physical meaning of what is found in the contrast variation method and demonstrates that it is the "surface-averaged" SLD of the scattering objects.

Comments: Previous studies by Thomas et al. 2014 applying the homogeneous law show results which are very similar and more importantly indicate that the acid digestion process used to isolate the kerogen studied alters the properties of the kerogen.

Responses: Since we do not have the samples from Thomas *et al.* published in 2014, we cannot know if there are heterogeneous surfaces in their materials. In fact, Thomas did not report any surface heterogeneity information in his paper at all as the homogeneous method cannot provide any heterogeneous surface information. Our GPSLM method provides the possibility to obtain the surface heterogeneity information (Δ_H^2) for the first time.

We have also applied our GPSLM method to estimate the possible errors of ρ_A and S_T if there are heterogeneous surfaces with nonzero Δ_H^2 (see the Supplementary Note 6 and Supplementary Figure 3 in the revised supplementary document). As mentioned already, the possible errors depend on the surface heterogeneity and the fluid SLD used to estimate ρ_A .

Thomas *et al.* 2014 paper is a nice and careful paper which focused on the effect of ρ_A . If we assume that the samples in the paper by Thomas *et al.* have similar values of Δ_H^2 as our kerogen samples, the method used by Thomas *et al.* actually gives similar values for ρ_A and slightly overestimates the

S_T compared with the GPSLM method. This in turn indicates that a new method/theory is urgently needed to estimate the surface heterogeneity as the traditional method sometimes may not be able to give even a hint of the surface heterogeneity. Our method not only can work for the isolated kerogens but also can be applied to shale rocks composed of both minerals and organic matters.

As for the sample preparation, we agree with the reviewer that the isolated kerogens might be different from the real kerogens in shale rocks. But even for the isolated kerogen materials, their surface heterogeneity has not previously been reported. We are planning to study the shale rocks with the new GPSLM method.

Comments: The author's interpretation of BET data is simplistic as we now know that sample preparation drastically affects the surface area measured in N₂ adsorption. Furthermore, inversion for pore dimensions typically requires the assumption of slits which is: 1) not true and 2) inconsistent with typical scattering models.

Responses: The BET surface area is the commonly used surface area characterization in the majority of the studies for porous systems. The isolated kerogens from the same batch were used for the BET and SANS measurements. As the samples were treated with the same way, we feel that it is reasonable to compare the results between BET and SANS measurements. We have revised the manuscript to show explicitly that the comparison is between the samples from the same batch.

But we agree with the reviewer that the properties of isolated kerogens may be different from that of the original kerogens in shale rocks as they depend on the sample preparation for kerogen isolation. But our method provides a way to study the surface heterogeneity of real shale rocks with kerogens imbedded inside and this is part of our future work.

As for the approximate pore size value mentioned in the manuscript, we agree with the reviewer that the estimation of pore size is complicated. But we are only interested in a rough estimation of the order of magnitude of the length scale at that Q range. We do not intend to imply that this is the exact pore size. That is why we give a range of length scale such as pore size larger than 200 Å, instead of giving single pore size value in the manuscript. We also want to point out that the accuracy of this estimation has no effect on the main results of our paper. The manuscript is revised to clearly address this point.

Comments: The authors make a claim in the discussion section that by using "GPSLM" they can quantify surface heterogeneity of kerogen in shale but as Thomas et al. (2014) claim, they may not be seeing the true kerogen response but the variability of the kerogen reaction to the acid digestion process.

Responses: As mentioned above already, we used a common contrast variation method similar to what Thomas reported in his paper. We agree with the reviewer that the isolated kerogens could be different from the real kerogens in shale rocks. But even for the isolated kerogen materials, their surface heterogeneity has not yet been reported in the literature. Our new GPSLM provides a way to measure the surface heterogeneity for the isolated kerogens. Future studies are needed to study the difference between isolated kerogens and the original kerogens in shale rocks. We are planning to study the shale rocks with the GPSLM method in near future.

Reviewers' comments:

Reviewer #2 (Remarks to the Author):

This reviewer appreciate authors including reviewers' recommendations and revise the manuscript in response of critiques. I hope that my efforts help authors improve the significance of their work and clarify main points.

For my comment 1, they addressed very well.

For my comment 2, authors are right about the contrast of the original Porod law, $\rho - \rho_{\text{solvent}}$. I might be confused about the intensity formula, for example equation 14 in the supporting information. There, the contrast should be $\rho_i - \rho_{\text{mean}}$. Otherwise, the intensity might keep getting higher even if one of compositions were far above 50%. For the Porod constant, however, as authors pointed out, it should be $\rho - \rho_{\text{solvent}}$. Since S will decrease for the extreme case. So Babinet principle works.

For my comment 3, appreciated.

For my comment 4, I appreciated authors for taking into account this. Apart from the fractal, I would like to again point out the two Porod regions in SAS curves, although authors claimed that their range does not cover the very small Q region. The work by Spalla et al. might be still useful because of authors' Figure 1. If authors measure only high Q part of SAS data, they might have missed the surface area of each big domains of S_1 , S_2 , and S_3 . Authors might have measured only the internal nanoscale pores. Thus, it is important that authors formula does not include these areas.

This argument does not claim that authors' work is wrong. I suggests that they need to draw a better boundary: what are the assumptions and limitations.

For comment 5, the table added much value.

Full derivations of equations should be included in the supporting information no matter requested or not.

When the issue related to the comment 4 is successfully addressed, I would be happy to suggest its publication.

Reviewer #3 (Remarks to the Author):

I think the work deserves to be published, just not in Nature. This is not a paper for the general readers of Nature. This is a sophisticated technique that a few specialists understand. This paper would benefit from publication in a more specialized journal where the authors can adequately expand on their generalization of the Porod method. The clarification of the preparation of BET and SANS samples did help a little.

Responses to Reviewers' Comments:

Reviewer #2 (Remarks to the Author):

This reviewer appreciate authors including reviewers' recommendations and revise the manuscript in response of critiques. I hope that my efforts help authors improve the significance of their work and clarify main points.

Responses: We thank the reviewer for his constructive comments.

For my comment 1, they addressed very well.

Responses: We thank the reviewer for agreeing with our responses.

For my comment 2, authors are right about the contrast of the original Porod law, $\rho - \rho_{\text{solvent}}$. I might be confused about the intensity formula, for example equation 14 in the supporting information. There, the contrast should be $\rho_i - \rho_{\text{mean}}$. Otherwise, the intensity might keep getting higher even if one of compositions were far above 50%. For the Porod constant, however, as authors pointed out, it should be $\rho - \rho_{\text{solvent}}$. Since S will decrease for the extreme case. So Babinet principle works.

Responses: We are glad that the reviewer agrees with us.

For my comment 3, appreciated.

For my comment 4, I appreciated authors for taking into account this. Apart from the fractal, I would like to again point out the two Porod regions in SAS curves, although authors claimed that their range does not cover the very small Q region. The work by Spalla et al. might be still useful because of authors' Figure 1. If authors measure only high Q part of SAS data, they might have missed the surface area of each big domains of S_1 , S_2 , and S_3 . Authors might have measured only the internal nanoscale pores. Thus, it is important that authors formula does not include these areas.

This argument does not claim that authors' work is wrong. I suggests that they need to draw a better boundary: what are the assumptions and limitations.

Responses: We thank the reviewer to call our attention to this interesting paper (Spalla, O., Lyonard, S. & Testard, F. Analysis of the small-angle intensity scattered by a porous and granular medium. *J. Appl. Crystallogr.* **36**, 338–347 (2003)). However, we would like to point out that the paper mentioned by the reviewer actually agrees with our interpretation. In fact, both our theory and that reported in the Spalla's paper show that if the SAS data are measured at sufficiently large Q region, as in our current case, the total surface area S_T extracted from these two methods include surface area of both big domains and internal nanoscale pores. So our theory does not miss the surface area of big domains even for the special cases mentioned in Spalla's paper. We have already included the paper by Spalla *et al.* in reference 28 of the manuscript.

It should be noted that for the special cases with two Porod law regions mentioned by Spalla's paper, the explanation of the surface area extracted from the high Q Porod law is well defined as it is due to the scattering of total surface area of a sample. However, the surface area extracted from the low Q Porod region is very ill defined. Even in the paper from Spalla *et al.*, they used a special term, "envelope surface", instead of the surface of grains, to describe the low Q Porod law region. Specifically, they mentioned in the paper that the envelope surface results from a smoothing of the outer shape of the grain, which means the envelope surface is not a true surface on a porous grain. Since we only consider high Q Porod law region in our current work, the definition is clear and without any ambiguity.

To avoid the confusion, Spalla *et al.* worked on special cases that the total surface area is dominated by the internal surface area so that they can use the total surface area to approximate the internal surface area in their paper. In addition, for the case with two Porod regions reported by Spalla *et al.*, they had to subtract the contribution of low Q Porod scattering to remove the contribution of inter-grain porosity (see section 3.1. of Spalla's paper) when calculating the "inner porosity" using the invariant method. This again shows that both the surface of inter-grain pores and inner-grain pores contribute to high Q region. Only when the big domain surface area is negligible, they can use high Q Porod scattering to approximately extract the specific surface area of inner pores (see Section 3.2. of Spalla's paper).

For comment 5, the table added much value.

Full derivations of equations should be included in the supporting information no matter requested or not.

Responses: The authors have already put the derivation of equations in previous supporting information. In the modified version of supporting information, we give more details and also add more steps into the derivation.

When the issue related to the comment 4 is successfully addressed, I would be happy to suggest its publication.

Responses: The authors hope the reviewer is satisfied with our responses to comment 4.

Reviewer #3 (Remarks to the Author):

I think the work deserves to be published, just not in Nature. This is not a paper for the general readers of Nature. This is a sophisticated technique that a few specialists understand. This paper would benefit from publication in a more specialized journal where the authors can adequately expand on their generalization of the Porod method. The clarification of the preparation of BET and SANS samples did help a little.

Responses: The authors respectfully disagree with the reviewer. Our paper provides a novel, generic, and non-invasive method to probe the variation of the surface properties of porous materials that is difficult for other experimental methods to obtain. In fact, there are very few methods, if any, that can extract the surface heterogeneity of inner pores buried deeply inside the bulk materials without using destructive sample preparation procedures. Both reviewer 1 and reviewer 2 have also agreed with our results. It should be noticed that the generalized Porod's scattering law method (GPSLM) can be used not only in heterogeneous porous materials, but also in many other heterogeneous materials such as colloidal suspensions and biological materials. Thus, GPSLM has high potential to draw general interest.

Reviewers' comments:

Reviewer #2 (Remarks to the Author):

I agree that the authors' method is technically sound as long as the following conditions are met.

1. Porous materials under investigation have completely open pores to methane.
2. Analysis at the highest Q is important not to miss any pore, which authors replied.
3. More importantly, authors' method is only well defined in the system where the domains of heterogeneities are not coupled or correlated. It is not defined nor discussed for a core-shell like system, for example a porous particle containing domains of various different densities. I am referring to H. Endo, Physica B. 2006, 385-386, page 682-684 and the cross terms in the partial structure factors at high Q. It is hard to imagine that natural system like kerogen would not have that type of heterogeneity.

Unfortunately, I cannot find their assumptions and limitations. This method may truly work for those cases, but with the materials provided here I cannot judge how general the method really is.

Minor comments regarding presentation are following. The authors emphasize that this technique is unique in measuring the heterogeneity of particulates. In the supporting information, ρ_M should be presented instead of square of ρ_M so that readers can see the variance right away. Not sure why Figure 4a y-label is not IR as opposed to Figure 2b. For sample 3 in Figure 4, not clear how they could determine ΔH without observing minima in IR. Did they fit IR with the quadratic equation? I would think then showing fits overlaid with the IR data in Figure 4 would make the analysis more convincing.

Responses to the comments by Reviewer #2:

Overall, we thank the reviewer for the constructive comments and we will address the points raised by the reviewer below.

Comments: I agree that the authors' method is technically sound as long as the following conditions are met.

1. Porous materials under investigation have completely open pores to methane.
2. Analysis at the highest Q is important not to miss any pore, which authors replied.

Responses: We are glad to know that the reviewer agrees that GPSLM method is technically sound.

Comments: 3. More importantly, authors' method is only well defined in the system where the domains of heterogeneities are not coupled or correlated. It is not defined nor discussed for a core-shell like system, for example a porous particle containing domains of various different densities. I am referring to H. Endo, Physica B. 2006, 385-386, page 682-684 and the cross terms in the partial structure factors at high Q . It is hard to imagine that natural system like kerogen would not have that type of heterogeneity.

Unfortunately, I cannot find their assumptions and limitations. This method may truly work for those cases, but with the materials provided here I cannot judge how general the method really is.

Responses: Our understanding is that the reviewer wants us to clearly state the assumptions and limitations of this method.

The GPSLM method should work fine with just three assumptions, which are also partly mentioned by the reviewer:

- 1) There is a clear Porod's law scattering region (i.e. the scattering intensity $I(Q) \sim Q^{-4}$) at relatively high Q region to include the scattering of all pore surfaces.
- 2) All pores are accessible to gas/liquids.
- 3) The SLD of the solid matrix does not change with the loading of gas/liquids.

We have now clearly stated the assumptions in the revised manuscript (page 8) and the supporting information (last paragraph of Supplementary Note 1).

The reviewer seems to feel that the GPSLM method may not work for highly correlated structures so a third assumption may be needed. The reviewer specifically mentioned the core-shell like system. We respectfully disagree with the reviewer on this.

The reviewer is concerned about the scattering of highly correlated structures. For the three assumptions listed above, only assumption 1 is related with the scattering feature from samples.

Thus, the reviewer's concern is only true if the Porod's scattering law does not work for highly correlated structures. However, this is definitely not true. Even for the special case of the core-shell structure, which is a highly correlated system, the Porod's scattering law still works fine.

The scattering of the core-shell model is well-known in the literature. Figure R1 shows the simulated scattering intensity from a system composed of core-shell particles after convoluting with a realistic common instrument resolution function obtained from NG-7 SANS instrument in NIST Center for Neutron Research (NCNR). It should be noticed that all the measured experimental SANS intensity is the result of theoretical intensity convoluting with an instrument resolution function. All parameters used for this core-shell model are given in Figure R1. It is clear that the exact total surface area S_T (known for the simulated system) and the experiment S_T (obtained from our GPSLM) are the same.

Figure R1. The simulated scattering intensity for a system composed of core-shell particles after convoluting with instrument resolution function. The parameters for the core-shell particles are given in the figure. R_c , R_s , ρ_c , ρ_s , and ρ_f are inner core radius, outer radius, core SLD, shell SLD, and fluid SLD, respectively. The fitting to the Porod's law scattering region gives the total surface area S_T (experiment value) the same as the S_T calculated directly from the geometry of this particle (exact value).

We can further prove theoretically that the Porod's scattering law works totally fine for the core-shell structure. In the core-shell system, the domains of heterogeneities, i.e. the core domain and

the shell domain, are highly coupled and correlated. We demonstrate briefly here that even in this highly correlated system, the equation of Porod's scattering law is still valid, which means that our GPSLM method works properly too from the scattering point of view.

For core-shell particles with inner core radius as R_c , outer radius as R_s , core SLD as ρ_c , and shell SLD as ρ_s , and immersed in fluid with SLD as ρ_f , the Fourier transformation of the particle can be written as

$$F_{core-shell}(Q, R_c, R_s, \rho_c, \rho_{in}, \rho_f) = F_c + F_s$$

$$(F_c = \left(\frac{4}{3}\pi R_c^3\right) (\rho_c - \rho_s) \frac{3j_1(QR_c)}{QR_c} \text{ and } F_s = \left(\frac{4}{3}\pi R_s^3\right) (\rho_s - \rho_f) \frac{3j_1(QR_s)}{QR_s} \text{ respectively.})$$

The core-shell intra-particle structure factor can be written as:

$$P_{core-shell}(Q, R_c, R_s, \rho_c, \rho_{in}, \rho_f) = |F_{core-shell}|^2 = F_c^2 + F_s^2 + F_c F_s^* + F_c^* F_s .$$

The scattering intensity of the core-shell system can be written as:

$$I(Q) = n P_{core-shell}(Q, R_c, R_s, \rho_c, \rho_{in}), \text{ where } n \text{ is number density of the core-shell particles.}$$

(For the simplicity reason, we can ignore the inter-particle structure factor here.)

$$F_c^2 \xrightarrow{Q \rightarrow \infty} 2\pi (\rho_c - \rho_s)^2 (4\pi R_c^2) \frac{1}{Q^4}, \text{ which gives the surface area of inner core.}$$

$$F_s^2 \xrightarrow{Q \rightarrow \infty} 2\pi (\rho_s - \rho_f)^2 (4\pi R_s^2) \frac{1}{Q^4}, \text{ which gives the surface area of outer shell.}$$

$F_c F_s^* + F_c^* F_s \xrightarrow{Q \rightarrow \infty} 0$ because the average intensity after convoluting with an instrument resolution function should be zero (a necessary step for all the measured scattering intensity).

This reduces to the general equation of the Porod's scattering law (equation (4) in the main text):

$$I(Q) \xrightarrow{Q \rightarrow \infty} 2\pi \left[\frac{1}{S_T} \int (\rho(S) - \rho_f)^2 dS \right] Q^{-4} \frac{S_T}{V} = 2\pi \langle \Delta\rho^2 \rangle_s Q^{-4} \frac{S_T}{V} = C_{GPS} Q^{-4}$$

The above derivation is an exact theoretical result without any assumption. The only requirement is that the Q value should be sufficiently large (i.e. the condition 2 mentioned by the reviewer) so that the thickness of the core-shell particle $t \gg \frac{1}{Q}$. Therefore, the GPSLM described in this manuscript is very general.

In summary, as long as the Porod's scattering law is valid, the GPLSM method should work properly if all pores are accessible and the SLD of solid matrix does not change. Hence, to search for special structures with smooth interface that the GPSLM method fails to work is equivalent to search for the cases in which the Porod's scattering law theory has to fail. As the Porod's scattering feature has been used for many decades, it would be very challenging to find a system with only smooth interfaces but without a Porod's scattering region. For a few highly correlated structures, including the core-shell structure discussed here, we all find that the Porod's scattering law works properly. It is certainly interesting to keep searching the validity of the Porod's theory for other types of highly correlated systems in future. However, we feel that this may be beyond the scope of this manuscript.

Comments: Minor comments regarding presentation are following. The authors emphasize that this technique is unique in measuring the heterogeneity of particulates. In the supporting information, rho_M should be presented instead of square of rho_M so that readers can see the variance right away.

Responses: the authors thank the reviewer for the suggestion. Because ρ_M^2 is the surface averaged square of scattering length density (SLD), i.e. $\rho_M^2 \equiv \frac{1}{S_T} \int \rho(S)^2 dS$, and the SLD of components can be either positive or negative, we present ρ_M^2 instead of ρ_M to include all the situations. However, in the kerogen samples, we know ρ_M must be positive and therefore we add a column of square root of ρ_M^2 , i.e. $\sqrt{\rho_M^2}$, in Supplementary Table 4 as suggested by the reviewer. The readers can now compare ρ_A and $\sqrt{\rho_M^2}$ and get an idea of heterogeneity of kerogens.

Comments: Not sure why Figure 4a y-label is not IR as opposed to Figure 2b.

Responses: the authors thank the reviewer for pointing out the difference. We have changed y-label of both Figure 2b and Figure 4a to $IR(\rho_f)$. In the revised manuscript and revised supporting information, we drop the Q dependence of $IR(Q, \rho_f)$ because $IR(Q, \rho_f) = \frac{C_{GPS}(\rho_f)}{C_{GPS}(\rho_f = 0)} = IR(\rho_f)$ in the Porod's scattering Q range and $IR(\rho_f)$ is actually independent of Q .

Comments: For sample 3 in Figure 4, not clear how they could determine delta_H without observing minima in IR. Did they fit IR with the quadratic equation? I would think then showing fits overlaid with the IR data in Figure 4 would make the analysis more convincing.

Responses: As mentioned in our previous version, we used the experimental $IR(\rho_f)$ value at the highest pressure to approximate the ΔH . The conclusion of the paper is not affected by this approximation. In the revised manuscript, we change the symbol of Sample 3 in Figure 4b to be open grey triangle, and add additional explanation in the revised manuscript (page 14 and figure caption of Figure 4b) and the revised supporting information (Supplementary Table 4).

REVIEWERS' COMMENTS:

Reviewer #2 (Remarks to the Author):

Glad that authors clarified and improved the presentation. Authors however should include their argument about the core-shell in the supporting info to clarify readers about their assumption $F_c F_s^* + F_c^* F_s \rightarrow 0$.

Koregen structures may not belong to "special structure" category in which their theory may fail, but certainly there can be such materials systems. Thus, I am not certain that "general" is a good naming for their method. It's authors' call.

Other than this, I do not have any more problem in this manuscript to be published.

Responses to the comments by Reviewer #2:

Comments: Glad that authors clarified and improved the presentation. Authors however should include their argument about the core-shell in the supporting info to clarify readers about their assumption $F_c F_s^* + F_c^* F_s \rightarrow 0$.

Responses: The authors thank the reviewer for all the suggestions that help us improve our manuscript. The argument about core-shell model is now put into Supplementary Note 1 and Supplementary Figure 1 in the revised Supporting Information.

Comments: Kerogen structures may not belong to "special structure" category in which their theory may fail, but certainly there can be such materials systems. Thus, I am not certain that "general" is a good naming for their method. It's authors' call.

Responses: The word “general” in the generalized Porod’s scattering law method (GPSLM) is to emphasize that GPSLM is a theory extending the original Porod’s scattering law to heterogeneous multi-phase systems. Therefore, we choose to maintain the original name of GPSLM. It is certainly an interesting idea to continue searching “special structures” that this theory may not apply. However, this is beyond the scope of the current manuscript.

Other than this, I do not have any more problem in this manuscript to be published.

Responses: The authors thank the reviewer to agree the publication of this manuscript.